# Prevalence of Culturable Bacteria and Yeasts in the Nasopharynx Microbiota during the Physiological Course of Pregnancy

**DOI:** 10.3390/jcm12134447

**Published:** 2023-07-02

**Authors:** Urszula Kosikowska, Dominik Franciszek Dłuski, Dorota Pietras-Ożga, Bożena Leszczyńska-Gorzelak, Sylwia Andrzejczuk

**Affiliations:** 1Department of Pharmaceutical Microbiology, Medical University of Lublin, 20-093 Lublin, Poland; urszula.kosikowska@umlub.pl; 2Department of Obstetrics and Perinatology, Medical University of Lublin, 20-954 Lublin, Poland; bozena.leszczynska-gorzelak@umlub.pl; 3Department of Epizootiology and Clinic of Infectious Diseases, University of Life Sciences in Lublin, 20-950 Lublin, Poland; dorota.ozga@gmail.com

**Keywords:** asymptomatic patients, bacterial colonization, yeast colonization, nasopharyngeal microbiota, opportunistic pathogens, pregnancy, MALDI-TOF MS

## Abstract

The aim of the study was to compare the prevalence of the nasopharyngeal carriage of culturable microorganisms in the microbiota of asymptomatic women with a physiological pregnancy (PW) and nonpregnant women (NPW). Nasopharyngeal swabs were collected from 53 PW and 30 NPW to detect bacterial and fungal colonization. Isolates were identified using the culture method and the MALDI-TOF MS technique. The nasopharyngeal microbiota (NPM) partially differed between PW and NPW. These differences in the frequency of nasopharyngeal colonization between the PW and NPW groups were not statistically significant (*p* > 0.05); all cases were colonized by bacteria and only two cases in the PW group were colonized by yeasts, namely, *Rhodotorula* spp. High levels of staphylococcal colonization, including predominantly coagulase-negative staphylococci and *S. aureus* in the nasopharyngeal sample, were present in both groups. The reduced number of Gram-negative rods colonized in the cases studied was seen in samples from the NPW group, particularly with *Enterobacterales*, and anaerobic *Cutibacterium* spp. were isolated only in the PW group (*p* < 0.05). Moreover, a higher carriage rate of *Enterobacter aerogenes* colonization was statistically significant (*p* < 0.05) and correlated with the NPW group. Pregnancy may disturb the composition of the NPM represented by commensals and opportunistic bacteria and promote yeast colonization as compared to nonpregnant women.

## 1. Introduction

The human nasopharynx is an appropriate environment for the life of various types of microorganisms, mainly bacteria, that may inhabit its cavity as the microbiota. Pregnancy is a special state that induces a number of changes in the maternal body. It is associated with major changes in hormonal regulation, metabolism, immunodeficiency, and physiological status [1,2,3,4,5]. The need for the immune system to tolerate the growth of the fetus results in an immunocompromised state of the woman, who may be susceptible to colonization by opportunistic pathogens and infections [6,7,8]. The composition of the maternal microbiota and its changes in different body parts and tissues, including the oral cavity and respiratory tract mucosa, have been shown to be an important factor in their functions, the regulation of the immune system, metabolism, and resistance to infectious diseases [5,8,9,10]. Microorganisms colonizing the amniotic fluid, digestive system, and respiratory tract of neonates have been extensively analyzed and well documented in several studies [5,6,7,8,9,10,11]. Microbial diversity and changes in microbiota composition are critical for pregnancy, probably through their effects on the maternal physiological, immunological, endocrine, and metabolic status. Furthermore, certain intestinal and oral bacteria, when presented as opportunistic pathogens, may have deleterious effects on maternal and infant health and the duration of pregnancy [10,12,13,14,15,16,17]. The way the respiratory microbiota changes during pregnancy and the way it is associated with various health conditions have become much more interesting because of its influence on pregnancy and future offspring.

The opportunistic microorganisms with their dual features of commensalism and pathogenicity are the reason why these bacteria and fungi are defined as pathobionts [18]. Their activity in the human body is an effect of changes in the healthy microbiota and of complex interactions of various genetic, exposure, microbial, and host factors that lead to their selection and expansion. For this reason, we identified and compared the nasopharyngeal colonization with common opportunistic pathogens as potential risk factors responsible for severe infectious diseases in the body of the hosts, healthy pregnant (PW) and healthy nonpregnant women (NPW). We evaluated this phenomenon by identifying mainly aerobic bacteria and fungi isolated from the nasopharynx of both groups. This study was conducted to analyze whether pregnancy had an effect on the colonization of the nasopharyngeal microbiota (NPM). A better understanding of how the changes in the maternal nasopharyngeal microbiota may underlie the susceptibility of pregnant women to opportunistic pathogen colonization may be helpful in preparing preventive interventions that may be useful in improving pregnancy outcomes. Some microorganisms colonizing the respiratory mucosa, including nasopharyngeal isolates, reflect the opportunistic strains currently circulating in the community.

## 2. Materials and Methods

### 2.1. Patients

Eighty-three asymptomatic female participants were enrolled in this study between December 2017 and September 2019 to determine the effect of pregnancy on nasopharyngeal colonization, mainly with aerobically growing bacteria, and the prevalence of opportunistic pathogens. The participants were divided into two groups: 53 pregnant women (PW group) and 30 non-pregnant women (NPW group—control group). The PW group with a physiological course of pregnancy between the 5th and 40th week of gestation, whose average age was 29.68 ± 4.63 (with a range from 19 to 41 years), was diagnosed at the Department of Obstetrics and Perinatology, Medical University of Lublin, Poland. In the NPW group, the mean age was 31.47 ± 8.64 years (with a range from 22 to 45 years).

After an accurate clinical data collection, the study inclusion and exclusion criteria were applied to the PW and NPW groups. The inclusion criteria in the PW group were: (a) singleton pregnancy, (b) absence of chronic disease, (c) absence of autoimmune disease, and (d) absence of infection. The exclusion criteria in the PW group were: (a) multiple pregnancy, (b) presence of chronic disease (diabetes, hypertension, hypothyroidism, hyperthyroidism, or other endocrine disease), (c) presence of autoimmune disease (e.g., antiphospholipid syndrome, lupus erythematosus), (d) presence of inherited thrombophilia, (e) presence of pregnancy-associated disease (e.g., HELLP—hemolysis, elevated liver enzymes and low platelets—syndrome, gestational cholestasis), (f) presence of adverse pregnancy outcomes in the patient’s history (stillbirth, premature birth, miscarriage, fetal growth restriction), and (g) presence of any type of infection. Inclusion criteria in the NPW group were a good health without autoimmune and chronic diseases and the absence of infection. None of them were treated with anticoagulants and antimicrobials. All of the participants signed an informed consent to participate in the study and completed the questionnaire. The protocol was approved by the institutional review board (KE-0254/59/2016, Ethics Committee of the Medical University of Lublin, Poland). In order to protect the confidentiality of the cases, no information that could identify PW and NPW women was included in the case record form.

### 2.2. Laboratory Procedures

A total of 83 nasopharyngeal samples were collected from both groups. Patients were advised not to eat, suck sweets, use any antimicrobials or other oral or nasal medications, or smoke cigarettes (e-cigarettes) for at least 2–3 h prior to the nasopharyngeal swab collection. Samples were collected using sterile cotton swabs with Amies medium (Copan Liquid Amies Elution Swab, eSwab, Copan, Brescia, Italy). In the PW group, samples were collected at the Department of the Obstetrics and Perinatology, Medical University of Lublin within 2–4 h of the patient’s admission and then transferred to the Department of Pharmaceutical Microbiology, Medical University of Lublin. In the NPW group, the samples were collected at the Department of Pharmaceutical Microbiology, Medical University of Lublin by medical microbiology specialists.

The components of the nasopharyngeal microbiota were first detected and identified using the culture method. A microbial examination of the samples was performed by inoculation onto agar media routinely used for the selective growth of microorganisms. Bacteria were cultured and isolated on appropriate nonselective media such as blood agar with 5% sheep blood for aerobic bacteria, *Haemophilus* chocolate agar (HAEM, bioMérieux, Craponne, France, for microaerophilic bacteria with high growth requirements), Wilkins Chalgren agar supplemented with sheep blood (for anaerobic bacteria, bioMérieux, Craponne, France), and Sabouraud dextrose agar (bioMérieux, Craponne, France) supplemented with chloramphenicol (for yeasts). After inoculation, the media were incubated at a temperature of 35 °C under aerobic conditions for up to 2–3 days for facultatively anaerobic bacteria and yeasts, under aerobic conditions with an elevated 5% CO_2_ concentration for microaerophilic bacteria, and for 4–6 days with the GENbag anaer atmosphere generator (bioMérieux, Craponne, France) for anaerobic bacteria. After incubation, morphologically distinct colonies growing on the agar media were selected and cultured on the agar media for bacteria and fungi. Isolates were classified according to phenotypic characteristics, including the growth morphology (e.g., colony shape and size, smooth or rough surface, texture, colony elevation), biochemical properties and Gram staining. Subsequently, isolates were identified according to Bucka-Kolendo et al. [19] for bacteria and Dhiman et al. [20] for yeasts by the MALDI-TOF MS (matrix-assisted laser desorption/ionization–time-of-flight mass spectrometry, ultrafleXtreme, Bruker, Billerica, Massachusetts, USA) technique based on protein profile at the Department of Epizootiology and Clinic of Infectious Diseases, University of Life Sciences in Lublin, Poland. A few colonies of fresh microbial cultures were transferred to 1.5 mL Eppendorf tubes containing 150 µL of ultrapure water and mixed by vortexing. Then, 450 µL of ethanol was added to the suspension, and the solution was vortexed and centrifuged (13,000× *g*, 5 min). The supernatant was removed, and the microbial pellet was suspended in 40 µL of 70% formic acid (Fluka Chemie GmbH, Buchs, Switzerland). After vortexing, 40 µL of acetonitrile (Fluka Chemie GmbH, Buchs, Switzerland) was added, and the sample was centrifuged (13,000× *g*, 2 min). For identification, 1 µL of protein extract was applied to the MALDI target plate and overlaid with the same volume of α-cyano-4-hydroxycinnamic acid (HCCA) solution (Bruker, Billerica, Massachusetts, USA). The mass spectra obtained for each isolate were processed using the MALDI Biotyper 3.1 database, Build (66) (Bruker, Billerica, Massachusetts, USA).

### 2.3. Statistical Analyses

A test sample with a group size of 53 (PW group) and 30 (NPW group) women, respectively, will reliably (with a probability greater than 0.9 for the test power) detect effect sizes d ≥ 0.747, using a two-sided detection criterion that allows a maximum Type I error rate of α = 0.05. Variables expressed at the ordinal or nominal level were analyzed using chi-square tests. When the conditions for using the chi-square test were not met, Fisher’s exact test was used. Parametric tests (Student *t*-test or ANOVA (analysis of variance)) or their nonparametric counterparts (Mann–Whitney U-test or Kruskal–Wallis test) were used to analyze the quantitative variables presented in groups. The choice of tests and the normal distribution of continuous variables were assessed using the Shapiro–Wilk test. The calculations were carried out in the statistical environment of R version 3.6.0, PSPP, and MS Office 2019. The values of the parameters are presented as minimum, maximum, and median values. Statistical significance was set at *p* < 0.05. Statistical analyses were performed by the company E-Statystyka (statystyka@e-statystyka.com.pl) (Rybnik, Poland).

## 3. Results

### 3.1. Baseline Features of Participants

The characteristics of the PW and NPW groups analyzed are shown in Table 1. Selected parameters including duration of the pregnancy (trimesters), parity, and CRP (C-reactive protein) values were collected from the PW group only.

The median age of all 83 cases was 30.32 ± 6.38 years, and it was similar in the two study groups. In the PW group (Table 1), 53 women with a physiological course of pregnancy between the 5th and 40th (median = 33) week of gestation and aged between 19 and 41 years old (mean age: 29.68 ± 4.63) were recruited. In this group, 12.0% of the CRP values were 10 mg/L or higher. The NPW (control) group consisted of 30 nonpregnant women of childbearing potential, aged between 22 and 45 years (mean age: 31.47 ± 8.64). In addition, the descriptive statistical results for the PW cases for variables such as duration of pregnancy (weeks of gestation) and CRP are presented in Table 1.

### 3.2. Patient Colonization with Opportunistic Microorganisms

A total of 83 women had their nasopharynx colonized by bacteria alone (81 women) or by bacteria and yeasts together (2 women). In the majority of cases, a uniform growth of bacterial colonies was observed, and sometimes, some bacteria appeared in predominant numbers. The differences in the frequency of colonization of the nasopharynx by different species of bacteria and fungi in the PW group compared to the NPW group were not statistically significant (*p* > 0.05). From one to six different species (median of two species) were isolated from each sample in both the PW and NPW groups (Figure 1). As shown in Figure 1, the colonization of a case with only one species was observed more frequently in the PW group than in the NPW group (12/53, 22.6% vs. 6/30, 20.0%). The lower percentage of cases with two different species colonizing the nasopharynx was observed in the PW group compared to the NPW group (21/53, 39.6% vs. 13/30, 43.3%). In the PW group, the nasopharynx of 8/53 (15.1%) women was colonized by five to six different species of microorganisms compared to one woman (1/30, 3.3%) in the NPW group who was colonized by six different species. The occurrence of pregnancy did not significantly (*p* > 0.05) differentiate the carriage of microbes, including bacterial groups and phenotypes (e.g., Gram stain effect or aerobic growth state), in the nasopharynx of each case in the PW and NPW groups (Appendix A).

Bacteria belonging to facultative anaerobes were mainly found in the nasopharyngeal samples of both PW and NPW groups (Table 2). Gram-positive bacteria colonized 50/53 (94.3%) of the PW cases and 27/30 (90.0%) of the NPW cases (*p* = 0.463). Notably, 48/53 (90.6%) vs. 27/30 (90.0%) cases in the PW and NPW groups, respectively, were colonized with different species of staphylococci. A number of cases colonized with Gram-negative bacteria belonging to the order *Enterobacterales* (three families: *Enterobacteriacae*, *Morganellaceae,* and *Erwiniaceae*) was observed both in the PW group (7/53, 13.2%) and in the NPW group (6/30, 20.0%) (Table 2). Fungi belonging to *Rhodotorula* spp. yeasts were isolated in the PW group in only two nasopharyngeal samples co-colonized with Gram-positive bacteria, mainly staphylococci (in both cases, *S. epidermidis*, and *S. haemolyticus* in one case) and *Corynebacterium pseudodiphtheriticum* in both samples.

The woman’s age was an important predictor of the occurrence of Gram-negative rods *Enterobacteriaceae* colonization in the nasopharyngeal cavity (Appendix A). Appendix A shows that in women aged 23.91 years, the odds ratio for *Enterobacteriaceae* was *P* = 0.21. The negative odds ratio indicates that the probability of occurrence of *Enterobacteriaceae* decreases by 21% with each year of age. Among women aged between 30.33 and 36.75 years, the odds ratio was much lower (*P* = 0.06 and *P* = 0.01, respectively). The remaining variables, including exposure to tobacco smoke, were not statistically significant predictors of the occurrence of these bacteria.

As shown in Table 3, Gram-positive cocci of coagulase-negative (CoNS) *S. epidermidis* and coagulase-positive (CoPS) *S. aureus* were the most abundant genera in the nasopharynx of the PW group (42/53, 79.2% and 14/30, 26.4%, respectively) and the NPW group (21/30, 70.0% and 7/30, 23.3%, respectively). The remaining CoNS *Staphylococcus* species were rarely isolated in both groups of patients.

According to Table 3, a difference in the occurrence of Gram-negative *Enterobacter aerogenes* (family *Enterobacteriacae*) colonization between the PW group (0/53, 0.0%) and the NPW group (3/30, 10.0%) was statistically significant (*p* = 0.019). No statistically significant relationship (*p* > 0.05) was observed between the pregnancy and the colonization of other bacterial and fungal species and their groups, genera, families, or orders.

The main bacterial colonization was staphylococci, which accounted for 75/83 (90.4%) of the nasopharyngeal samples. One to five *Staphylococcus* species were isolated from each case in the PW group and from one to three of them in the NPW group (Table 4).

It was also investigated in a total number of cases (*n* = 83) whether factors such as age, health status, place of residence, and exposure to tobacco smoke (active or passive) could be important predictors and adequately explain the probability of occurrence of the nasopharyngeal colonization with specific groups of bacteria and yeasts in the entire study sample (Appendix A). The statistically significant (*p* < 0.05) difference in colonization was observed with women positive for anaerobically grown Gram-positive bacilli of *Cutibacterium* spp. (formerly *Propionibacterium* spp.) or aerobically grown Gram-negative rods of the *Enterobacteriaceae* family in both groups (Appendix A). The predictors explained 36% and 17% of the variation in the probability of occurrence of *Cutibacterium* spp. (R^2^ = 0.36) and *Enterobacteriaceae* (R^2^ = 0.17) colonization, respectively. The results for other groups of microorganisms were not statistically significant (*p* > 0.05).

In the PW group, the gestational age and CRP level were not important predictors (*p* > 0.05) of the occurrence of colonization by specific groups, families, genera or order, and species of bacteria and fungi.

## 4. Discussion

Pregnancy is a remarkable process involving various changes in many systems, including the composition of the microbiota. It is well documented that the female body undergoes a number of physiological, hormonal, metabolic, and immunological changes during pregnancy [21,22]. The nasopharyngeal microbiota introduces the entire pool of symbiotic and commensal microorganisms, including opportunistic pathogens. It is also a highly diverse and very important ecosystem for health security [22,23,24,25]. It can also be extremely variable both within different parts of the body in a single case and among various individuals [26,27,28]. Pregnant women with the presence of opportunistic pathogens have a higher risk of infectious diseases or preterm labor and vertical transmission of various microorganisms, including bacteria and viruses, to their babies [29,30].

Our research showed that the nasopharyngeal microbiota of healthy patients varied according to pregnancy, without reaching statistical significance. The highest microbial species richness was associated with the pregnant group, although no statistical significance was achieved. In some cases, including *E. aerogenes* colonization in nonpregnant women (*p* = 0.019), other individual factors, such as age, may have a partial effect on the composition of the nasopharyngeal microbiota.

It has been documented that estrogen and progesterone during pregnancy can influence the gut and oral microbiota and create conditions for opportunistic infections [7,8,21,22,31]. In addition, physiological changes and differences in metabolic status during pregnancy increase the susceptibility to various oral diseases such as gingivitis and periodontitis [3,15,32]. According to Fujiwara et al. [9], the changes in microbial composition may be natural consequence of a physiological pregnancy. Some bacteria may be harmless colonizing microorganisms or highly invasive opportunistic pathogens (pathobionts) [18,33].

The differences in the composition of the nasopharyngeal microbiota of pregnant women compared to nonpregnant women have been reported during our research. According to our results, the microbial composition in the nasopharynx during pregnancy was dominated by members of the genera *Staphylococcus*, *Streptococcus* (e.g., *S. pneumoniae*) and *Corynebacterium*, and *Staphylococcus* species, including coagulase-positive (CoPS) *S. aureus* and the diverse group of coagulase negative staphylococci (CoNS), were frequently isolated as the PW and NPW groups of nasopharyngeal colonizers. The heterogeneous group of coagulase-negative staphylococci have historically been classified as nonpathogenic bacteria and are less frequently involved in clinically manifest infections [34,35]. Nowadays, due to patient- and diagnosis-related changes, CoNS represent one of the most important groups of nosocomial opportunistic pathogens, with *S. epidermidis* and *S. haemolyticus* as the most important species [34,36,37,38]. Various CoNS species can be found as major colonizers on the surface of the mucous membranes of the respiratory and urogenital mucosa [35,39,40]. *S. epidermidis* and other CoNS have become one of the most important causes of nosocomial infections in recent years [37,38,41,42]. In the study by Fujiwara et al. [9], in which the composition of the oral microbiota was determined both in women at different stages of pregnancy (early—7–16 weeks of gestation, mid—17–28 weeks, and late—29–39 weeks), and in healthy nonpregnant women, the genus *Streptococcus* was found to be the most abundant, followed by *Staphylococcus* spp. [43]. In contrast, only commensal streptococci were found in our study. Group A streptococci were not isolated. Group A *Streptococcus* (GAS) infections are known to be a major cause of maternal and infant mortality worldwide. According to known data sources, GAS colonization of the genital tract and sepsis-related maternal or infant mortality or morbidity are major problems [44]. It is also known that *Staphylococcus* spp. is predominant in the nose and nasopharynx of 1.5-month-old infants [43]. Moreover, as with other nosocomial pathogens, both CoNS and *S. aureus* are associated with increasing rates of antibiotic resistance and limited therapeutic options.

*S. aureus* has long been known to cause a wide range of infections, from mild to fatal [45]. According to a recent study [46], mothers who were colonized with *S. aureus* during the third trimester of pregnancy or at the time of delivery were more likely to have infants carrying these bacteria. In addition, *S. aureus* colonization (including MRSA—methicillin resistant strains) was extremely common in this cohort of mother–infant pairs. Infants born to mothers with *S. aureus* were more likely to be colonized, and early postnatal acquisition appeared to be a very important and primary mechanism. In addition to this, strains of this pathogenic species often cause infections by releasing toxins, with some proteins as virulence factors, including coagulase [47]. Among the diseases commonly seen in newborn babies, one of the most important toxic infections is known as SSSS (staphylococcal scalded skin syndrome). Symptoms of this disease include prominent blistering and exfoliation of the superficial skin surface, followed by the exfoliative toxins released by *S. aureus*.

Many studies have investigated the differences in various oral microorganisms in pregnant women as compared to nonpregnant women [9,48,49]. These changes may also affect the composition of the nasopharyngeal microbiota in these two groups of women. One example is Gram-positive cocci of the genus *Kocuria*, which belongs to the family *Micrococcaceae* which also includes morphologically similar *Staphylococcus* and *Micrococcus* species [50,51]. These strictly aerobic coccoids are commonly found in the microbiota composition of the skin and oral cavity of humans and animals. They are opportunistic pathogens and rarely cause infections. The first known clinical cases of *Kocuria rhizophila* infection caused sepsis and acute pancreatitis in a six-year-old boy [52] and a three-year-old girl [53]. *K. rhizophila* was described as a pathogenic factor only in the immunocompromised patients [52,53]. Currently, data on infectious diseases with *Kocuria* species show an increasing trend and include, e.g., urinary tract infections, catheter-associated bacteremia, keratitis, native valve endocarditis, peritonitis, descending necrotizing mediastinitis, brain abscess, and meningitis [54].

As was shown in our study, the difference in the colonization of the PW and NPW groups with anaerobically grown Gram-positive bacilli of *Cutibacterium* spp. (formerly *Propionibacterium*) or aerobically grown Gram-negative rods of the *Enterobacteriaceae* family was statistically significant (*p* < 0.05). The *Cutibacterium* species are nonsporulating bacilli and belong to the skin coryneform commensal group, the most studied species being *C. acnes* and *C. avidum* [55]. However, this bacterium can also cause a wide variety of infections, including endocarditis, postoperative shoulder infections, and neurosurgical shunt infections [55,56,57,58,59].

Based on our data, only in the PW group were the most common genera among the Gram-negative bacteria *Enterobacteriaceae* (including two *Klebsiella* species). In contrast, the presence of *E. aerogenes* was important and statistically significant (*p* < 0.05) only in the control group (NPW). Most authors refer to bacteria from the *Enterobacteriaceae* family, as they are mainly found in the gastrointestinal tract. Their presence as colonizers in our study may indicate potential risks for pregnant women. As shown by other authors, Gram-negative bacteria, mainly catalase-positive and oxidase-negative members of the *Enterobacteriaceae* family, are opportunistic pathogens responsible for the majority of nosocomial infections [60]. The presence of enterobacterial species commonly found in nasopharyngeal samples from pregnant women is important for their medical significance as opportunistic pathogens. According to the classification of Adeolu et al. [61], in our study, Gram-negative bacilli in the PW group were mainly represented by the *Enterobacterales* order with representative bacteria from the family *Enterobacteriaceae* (*Citrobacter* spp., *Enterobacter* spp. *Escherichia coli*, *Klebsiella* spp.), *Erwiniaceae* (*Pantoea* spp.), and *Morganellaceae* (*Proteus* spp.). Based on the results of the protein profiles, the presence of two *Klebsiella* species was detected, including *K. variicola*. This bacterium is known to cause severe bloodstream infections, to produce hypervirulent strains, and to exhibit multidrug resistance [62]. We found *Pseudomonadaceae* (including *P. aeruginosa*) as one of the most abundant genera in the PW group, which is consistent with other findings [7,63] and is often considered an etiological factor in chronic periodontitis in the oral microbiota [64]. In our results, two species of *Rhodotorula* yeasts were also identified only in the PW group, but this was not statistically significant. These yeasts were also found to co-occur with staphylococci. There are few publications available describing the contribution of *Rhodotorula* fungi to the upper respiratory tract microbiota. Recent mycobiome studies, for example, found a higher fungal diversity in caries-free children, with the most abundant taxa being *Rhodotorula mucilaginosa* [65]. Reports of fungal involvement in the oral microbiota [65,66] may suggest that studies to establish oral fungal communities are still needed, and that there is still much to discover about the oral mycobiome.

In our study, *Pantoea agglomerans* was found in only two women in the PW group. These bacteria can colonize the human body mainly through wound infection with plant and food material [67]. One reason for clinical and hospital-acquired infection with a *P. agglomerans* etiology may be an exposure of patients to medical devices or fluids contaminated with this bacterium during their hospitalization. Furthermore, the reason for a clinical infection or nosocomial sepsis with a *P. agglomerans* etiology may be the exposure of often immunocompromised patients to medical equipment or fluids contaminated with this bacterium [67]. In addition, the abundance of several bacterial taxa such as *Lactobacillus*, *Bifidobacterium*, *Streptococcus,* and *Escherichia coli* has been found to change during pregnancy [5,68]. Furthermore, microbial changes occurring during pregnancy can be considered a natural consequence of a healthy pregnancy. However, imbalances in the oral microbial composition and poor oral health have been shown to predispose pregnant women to a higher risk of developing periodontal disease [8,69]. The composition of the nasopharyngeal microbiota and maternal serum parameters, including CRP, and some factors such as active and passive smoking and the place of residence are associated with a higher risk of health problems and opportunistic infections in pregnant women.

There were no statistically significant differences between opportunistic bacterial and yeast colonization according to exposure to tobacco smoke or the place of residence. We had a small group of women in both pregnant and nonpregnant groups who were active or passive smokers and nonsmokers. This problem should be investigated in the next study. Tobacco smoke contains toxic substances that accumulate at various sites and induce changes in human health and environmental conditions [70]. According to Paropkari et al. [71], the influence of each environmental condition and its perturbations superimposed on the microbiota is unique and its sum is greater than its parts.

To our knowledge, this is the first study on the nasopharyngeal microbiota of healthy and uninfected women during pregnancy compared to nonpregnant women in good health. A similar issue was addressed by Crovetto et al. [67]. Their results showed differences between pregnant and nonpregnant women. They revealed significant differences in the overall composition of the nasopharyngeal microbiota in pregnant women with active infection compared to postinfection. They showed for the first time that the nasopharyngeal microbiota profile of pregnant women differed from previous cases, with a higher relative abundance of the Tenericutes and Bacteroidetes phyla and a higher abundance of the *Prevotellaceae* family.

Many studies have shown that pregnancy induces changes and differences in the gut microbiota (also referred to as fecal microbiota), especially in late pregnancy and rarely from the first to the third trimester [68,69,70,71]. In addition, studies of the differences in the gut microbiota of pregnant or nonpregnant women include, e.g., host metabolic and immune functions and some parameters such as serum C-reactive protein (CRP) values [72,73]. We have no knowledge of the influence or association of the trimester of pregnancy or CRP level with the nasopharyngeal microbiota composition and colonization of culturable microorganisms in healthy pregnant and nonpregnant women. Our research showed that parity or maternal serum CRP levels had no statistically significant association with nasopharyngeal colonization by culturable bacteria during pregnancy. Studies by some researchers have shown that the total viable oral microbial count was higher in all stages of pregnancy, especially in the first trimester of pregnancy compared with nonpregnant women [9]. As was shown in our research, the median CRP values for women in the PW group ranged from 0.04 to 13.83 mg/L, with 88% of the values being 10 mg/L or lower. This parameter value should be studied extensively in the diagnosis of infection in pregnancies with preterm labor or a preterm rupture of membranes. Understanding the time course of the increase in CRP values in pregnancy may help to clarify the appropriate use of this parameter in physiological and complicated pregnancies. As was shown by Sorokin et al. [74], high maternal CRP and interleukin-6 (IL-6) concentrations were associated with an increased risk of preterm birth (PTB). Furthermore, elevated concentrations of CRP, IL-6, and matrix metalloproteinase-9 (MMP-9) in the fetal and neonatal compartments were associated with a PTB risk and/or neonatal morbidity. Dodds et al. [75] found that the highest maternal CRP (≥8 mg/L) identified a subgroup of patients with the highest risk of preterm birth. It was found that patients with a prepartal CRP level of 10 mg/L or higher were found to have more febrile complications and subclinical intrauterine infection than women with normal CRP values [76,77]. Cohen et al. [68] found that the higher levels of CRP in viable intrauterine pregnancies (7.67 ± 1.81 mg/L) compared to those in ectopic pregnancies (2.43 ± 0.61 mg/L). Wirestram et al. [78] analyzed the CRP levels during normal pregnancy. The results showed that CRP was significantly higher in the third trimester compared to the first (*p* = 0.0006) and second (*p* < 0.0001) trimesters. Circulating high-sensitivity CRP (hs-CRP) is a widely accepted marker of chronic low-grade systemic inflammation [79], and it has been associated with heart disease [80], colorectal cancer [81], and complications of diabetes and obesity [82]. Fink et al. [83] demonstrated that maternal asthma, eczema, allergy, dog or cat in the household, pre-eclampsia, and parity were not associated with high-sensitivity CRP (hs-CRP) levels, but hs-CRP levels were three times higher at one week postpartum than at 24th weeks of gestation and were associated with the mode of delivery.

A number of limitations have made us unable to answer questions about the effect of pregnancy, its trimester, and some parameters that are often higher in pregnant women (e.g., CRP) on the composition and changes of the nasopharyngeal microbiota and the occurrence of various microorganisms in the nasopharynx during the course of a physiological pregnancy. The differences can be related to relatively a small number of pregnant and nonpregnant women groups, different populations, pets, place of residence, and different parameters which require several studies. Among these limitations, the relatively small specimen size did not allow us to draw proper conclusions from various subgroup comparisons; additionally, as there were no data on the upper respiratory tract microbiota during pregnancy, it was not possible to discern if differences were due only to the pregnancy itself. For this reason, future investigations are warranted to compare these and many more data and their effect on nasopharyngeal microbiota composition and differences among pregnant and nonpregnant women of the same age and health status.

## 5. Conclusions

This study demonstrated the differences in the composition of the nasopharyngeal microbiota in pregnant and nonpregnant women, with a particular emphasis on *Staphylococcus aureus* and coagulase-negative staphylococci, as well as *Cutibacterium* spp. and *Enterobacterales* colonization. The data may suggest that pregnancy can promote the differentiation of microorganisms in the nasal cavity and facilitate the colonization of opportunistic bacteria and yeasts, which requires confirmation with a larger study in a much larger group of patients, both pregnant and nonpregnant. Chronic diseases, including metabolic diseases (e.g., diabetes), pharmacotherapy used, frequency of infections of various types, physical activity, contact with animals, place of residence, and a number of other endo- and exogenous factors, including stress, should be taken into account. Further studies are needed to clarify the relationship between risk factors (e.g., health status, smoking), levels of selected diagnostic parameters (e.g., CRP) and the composition of the respiratory microbiota in pregnant women. In our opinion, the presence of specific groups of microorganisms in the respiratory microbiota has no direct clinical significance in healthy individuals. However, changes that occur, for example, during pregnancy or childbirth may be associated with the risk of opportunistic infections, including those associated with a change in the localization of microorganisms. One of the causes of many chronic and recurrent diseases and their associated risks is the disruption of the microbiota in different parts of the human body. Due to the paucity of literature on this subject, we do not know the full consequences of the presence of certain commensal microorganisms in the microbiota, the disturbances in the qualitative–quantitative composition, and the possible consequences for the mother and the newborn. The prevalence and characteristics of opportunistic pathogens can help in the prevention and treatment of infectious diseases. In our opinion, the idea and the methodological design show promising potential, and it is an important question because of the potential for diagnostic and therapeutic applications that could be derived. This is all the more reason why we believe that knowledge on this subject, especially in the case of pregnant women who undergo many changes, e.g., hormonal changes, is very important and is part of the new trends in health care and personalized medicine.

## Figures and Tables

**Figure 1 jcm-12-04447-f001:**
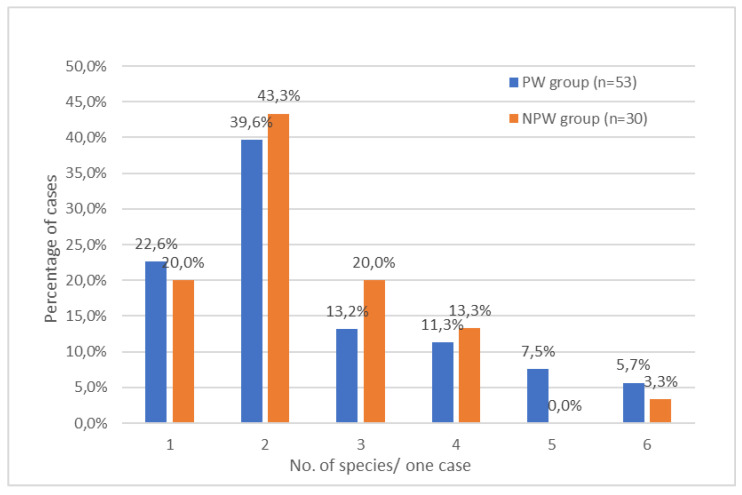
Percentage of species colonization prevalence in the nasopharynx cavity of each case in pregnant women (PW group) and nonpregnant women (NPW group).

**Table 1 jcm-12-04447-t001:** Characteristics of pregnant (PW group) and nonpregnant (NPW group) women.

No. (%) of Participants (*n* = 83)					
Parameters				PW Group (*n* = 53)	NPW Group (*n* = 30)
Age (years)	19–45			53 (100.0)	30 (100.0)
	Age range		19–35	48 (90.6)	19 (63.3)
			36–45	5 (9.4)	11 (36.7)
	M ± SD			29.68 ± 4.63	31.47 ± 8.64
	*Min–Max* (*Me*)			19.0–41.0 (30.0)	22.0–45.0 (29.0)
Place of residence	Village			21 (39.6)	5 (16.7)
	City			31 (58.5)	25 (83.3)
	No information ^(a)^			1 (1.9)	0 (0.0)
Smoking status					
Smoking	Never			33 (68.8)	19 (79.2)
	In the past			13 (27.1)	1 (4.2)
	Current			2 (4.2)	4 (16.7)
	No information ^(a)^			5 (9.4)	6 (20.0)
Passive smoking	No			44 (83.0)	20 (66.7)
	Yes			9 (17.0)	10 (33.3)
Pets	No			26 (49.1)	15 (50.0)
	Yes			27 (50.9)	15 (50.0)
Pregnancy trimester	First			5 (9.4)	
	Second			10 (18.9)	
	Third			38 (71.7)	
Weeks of gestation	M ± SD			29.30 ± 9.70	
	*Min–Max* (*Me*)			5.0–40.0 (33.0)	
Pregnancy	First			20 (37.7)	
	Second			14 (26.4)	
	Third			14 (26.4)	
	Fourth			5 (9.4)	
CRP (mg/L) ^(b)^ (*n* = 50)	CRP range	0–5		30 (60.0)	
		5.1–10		14 (28.0)	
		> 10		6 (12.0)	
	M ± SD			4.64 ± 3.87	
	*Min–Max* (*Me*)			0.04–13.83 (3.54)	

^(a)^ Persons did not add any information about a place of residence or smoking status; ^(b)^ CRP range was detected in 50 cases in the PW group; CRP reference value, 5 mg/mL. Abbreviations: CRP: C-reactive protein; *M:* mean; *Min:* minimum; *Max:* maximum; *Me*: median; *SD:* standard deviation.

**Table 2 jcm-12-04447-t002:** Frequency of nasopharynx colonization by selected groups of bacteria and yeasts in PW group and NPW group.

Microbial Colonizers	Number of Colonized Cases (*n* = 83)		
	PW Group (*n* = 53)	NPW Group (*n* = 30)	*P*
**Frequency of case colonized with various Gram-stain phenotypes of microorganisms**			
Only Gram-positive bacteria	38 (71.7)	19 (63.3)	0.467
Only Gram-negative bacteria	3 (5.6)	3 (10.0)	0.662
Gram-positive and Gram-negative bacteria	10 (18.9)	8 (26.7)	0.420
Gram-positive bacteria and yeasts*	2 (3.8)	0 (0.0)	0.533
**Frequency of case colonized with selected groups of opportunistic pathogens**			
**Gram-positive bacteria**			
Family: *Staphylococcaceae*	48 (90.6)	27 (90.0)	1.000
Family: *Streptococcaceae*	2 (3.8)	4 (13.3)	0.182
Other Gram-positive	26 (49.1)	14 (46.7)	1.000
**Gram-negative bacteria**			
Order: *Enterobacterales*	7 (13.2)	6 (20.0)	0.532
*Enterobacteriaceae*	4 (7.5)	5 (16.7)	0.273
*Morganellaceae*	1 (1.9)	0 (0.0)	1.000
*Erwiniaceae*	2 (3.8)	1 (3.3)	1.000
Order: *Pseudomonadales*	4 (7.5)	2 (6.7)	1.000
Other Gram-negative	5 (9.4)	2 (6.7)	1.000
**Yeasts**			
*Rhodotorula* spp.	2 (3.8)	0 (0.0)	0.146

* PW cases with co-colonization.

**Table 3 jcm-12-04447-t003:** Bacteria and fungi species colonizing the nasopharynx of cases in the pregnant (PW) and nonpregnant (NPW) group.

Family	Species	Number (%) of Colonized Cases		*χ^2^*	*p*
		PW Group (*n* = 53)	NPW Group (*n* = 30)		
**Gram-positive bacteria**					
*Staphylococcaceae*	*Staphylococcus aureus*	14 (26.4)	7 (23.3)	*0.096*	*0.756*
	*Staphylococcus epidermidis*	42 (79.2)	21 (70.0)	*0.895*	*0.344*
	*Staphylococcus haemolyticus*	3 (5.7)	4 (13.3)	*1.460*	*0.227*
	*Staphylococcus warneri*	3 (5.7)	4 (13.3)	*1.460*	*0.227*
	*Staphylococcus hominis*	6 (11.3)	1 (3.3)	*1.583*	*0.208*
	*Staphylococcus pasteuri*	1 (1.9)	1 (3.3)	*0.170*	*0.680*
	*Staphylococcus lugduensis*	0 (0.0)	1 (3.3)	*1.788*	*0.181*
	*Staphylococcus capitis*	2 (3.8)	0 (0.0)	*1.160*	*0.281*
	*Staphylococcus cohnii*	1 (1.9)	0 (0.0)	*0.573*	*0.449*
	*Staphylococcus saprophyticus*	3 (5.7)	0 (0.0)	*1.762*	*0.184*
	*Staphylococcus simulans*	1 (1.9)	0 (0.0)	*0.573*	*0.449*
	*Gemella haemolysans*	1 (1.9)	0 (0.0)	*0.573*	*0.449*
*Streptococcaceae*	*Streptococcus pneumoniae*	1 (1.9)	0 (0.0)	*0.573*	*0.449*
	*Streptococcus mitis*	1 (1.9)	0 (0.0)	*0.573*	*0.449*
	*Streptococcus parasanguinis*	1 (1.9)	3 (10.0)	*2.749*	*0.097*
	*Streptococcus salivarius*	0 (0.0)	1 (3.3)	*1.788*	*0.181*
*Enterococcaceae*	*Enterococcus faecalis*	1 (1.9)	1 (3.3)	*0.170*	*0.680*
*Micrococcaceae*	*Micrococcus luteus*	4 (7.5)	2 (6.7)	*0.022*	*0.882*
	*Kocuria rhizophila*	1 (1.9)	0 (0.0)	*0.573*	*0.449*
	*Kocuria palustris*	0 (0.0)	1 (3.3)	*1.788*	*0.181*
*Corynebacteriaceae*	*Corynebacterium pseudodiphtheriticum*	2 (3.8)	1 (3.3)	*0.011*	*0.918*
	*Corynebacterium accolens*	8 (15.1)	7 (23.3)	*0.878*	*0.349*
	*Corynebacterium tuberculostearicum*	3 (5.7)	3 (10.0)	*0.538*	*0.463*
	*Corynebacterium propinquum*	3 (5.7)	1 (3.3)	*0.226*	*0.634*
	*Corynebacterium amycolatum*	2 (3.8)	0 (0.0)	*1.160*	*0.281*
*Bacillaceae*	*Bacillus cereus*	8 (15.1)	1 (3.3)	*2.741*	*0.098*
	*Bacillus pumilus*	2 (3.8)	0 (0.0)	*1.16*	*0.281*
*Lactobacillaceae*	*Lactobacillus salivarius*	0 (0.0)	1 (3.3)	*1.788*	*0.181*
*Paenibacillaceae*	*Paenibacillus* spp.	1 (1.9)	0 (0.0)	*0.573*	*0.449*
*Propionibacteriaceae*	*Cutibacterium granulosum*	1 (1.9)	0 (0.0)	*0.573*	*0.449*
	*Cutibacterium avidum*	2 (3.8)	0 (0.0)	*1.160*	*0.281*
	*Cutibacterium acnes*	1 (1.9)	0 (0.0)	*0.573*	*0.449*
**Gram-negative bacteria**					
*Enterobacteriaceae*	*Citrobacter koseri*	1 (1.9)	1 (3.3)	*0.170*	*0.680*
	*Escherichia coli*	1 (1.9)	1 (3.3)	*0.170*	*0.680*
	*Klebsiella variicola*	1 (1.9)	0 (0.0)	*0.584*	*0.445*
	*Klebsiella oxytoca*	1 (1.9)	0 (0.0)	*0.573*	*0.449*
	*Enterobacter aerogenes*	0 (0.0)	3 (10.0)	*5.499*	*0.019 **
	*Raoultella ornithinolytica*	0 (0.0)	1 (3.3)	*1.788*	*0.181*
	*Pantoea agglomerans*	2 (3.8)	0 (0.0)	*1.160*	*0.281*
	*Pantoea septica*	0 (0.0)	1 (3.3)	*1.788*	*0.181*
*Morganellaceae*	*Proteus mirabilis*	1 (1.9)	0 (0.0)	*0.573*	*0.449*
*Xantomonadaceae*	*Stenotrophomonas maltophilia*	1 (1.9)	1 (3.3)	*0.170*	*0.680*
*Pseudomonadaceae*	*Pseudomonas aeruginosa*	1 (1.9)	0 (0.0)	*0.573*	*0.449*
	*Pseudomonas congelans*	1 (1.9)	0 (0.0)	*0.573*	*0.449*
	*Pseudomonas* spp.	1 (1.9)	0 (0.0)	*0.584*	*0.445*
*Pasteurellaceae*	*Haemophilus influenzae*	1 (1.9)	0 (0.0)	*0.573*	*0.449*
	*Haemophilus parainfluenzae*	3 (5.7)	1 (3.3)	*0.226*	*0.634*
*Moraxellaceae*	*Acinetobacter tandoii*	0 (0.0)	1 (3.3)	*1.788*	*0.181*
	*Moraxella_sg_Branhamella catarrhalis*	1 (1.9)	0 (0.0)	*0.573*	*0.449*
*Neisseriaceae*	*Neisseria mucosa*	0 (0.0)	1 (3.3)	*1.788*	*0.181*
**Yeasts**					
*Sporidiobolaceae*	*Rhodotorula mucilaginosa*	1 (1.9)	0 (0.0)	*0.573*	*0.449*
	*Rhodotorula minuta*	1 (1.9)	0 (0.0)	*0.573*	*0.449*

*χ^2^*—test statistics; *N*—counts; *p*—statistical significance.

**Table 4 jcm-12-04447-t004:** Number of *Staphylococcus* species colonizing the nasopharynx cavity of each case of pregnant (PW group) and nonpregnant (NPW group) women.

Group	Number of *Staphylococcus* Species per Sample	*Staphylococcus* Species											Number (%) of Colonized Cases
		CoPS	CoNS										
		*S. aureus*	*S. epidermidis*	*S. hominis*	*S. warneri*	*S. capitis*	*S. saprophyticus*	*S. haemolyticus*	*S. pasteuri*	*S. simulans*	*S. cohnii*	*S. lugduensis*	
PW (*n* = 53)	1		-	-	-	-	-	-	-	-	-	-	5 (9.4)
		-		-	-	-	-	-	-	-	-	-	22 (41.5)
	2			-	-	-	-	-	-	-	-	-	6 (11.3)
		-			-	-	-	-	-	-	-	-	4 (7.5)
		-		-	-	-		-	-	-	-	-	1 (1.9)
		-		-	-		-	-	-	-	-	-	2 (3.8)
		-		-		-	-	-	-	-	-	-	2 (3.8)
		-	-			-	-	-	-	-	-	-	1 (1.9)
	3			-	-	-	-		-	-	-	-	1 (1.9)
				-	-		-	-	-	-	-	-	1 (1.9)
		-				-	-	-	-	-	-	-	1 (1.9)
		-		-	-	-	-			-	-	-	1 (1.9)
	5			-	-	-	-		-			-	1 (1.9)
NPW (*n* = 30)	1		-	-	-	-	-	-	-	-	-	-	2 (6.7)
		-	-	-	-	-	-	-	-	-	-	-	9 (30.0)
		-	-	-		-	-	-	-	-	-	-	2 (6.7)
		-	-	-	-	-	-	-	-	-	-		1 (3.3)
		-	-	-	-	-	-		-	-	-	-	1 (3.3)
	2			-	-	-	-	-	-	-	-	-	2 (6.7)
		-			-	-	-	-	-	-	-	-	1 (3.3)
		-		-		-	-	-	-	-	-	-	2 (6.7)
		-		-	-	-	-		-	-	-	-	3 (10.0)
			-	-	-	-	-		-	-	-	-	1 (3.3)
		-		-	-	-	-	-		-	-	-	1 (3.3)
	3			-		-	-	-	-	-	-	-	2 (6.7)

Abbreviations: CoPS, coagulase-positive *Staphylococcus*; CoNS, coagulase-negative *Staphylococcus*; (-) sample without *Staphylococcus* spp. isolation; colours in cells—positive detections of a specific *Staphylococcus* species; red—*S. aureus*, blue—*S. epidermidis*, orange—*S. hominis*, light green—*S. warneri*, grey—S. capitis, purple—*S. saprophyticus*, green—*S. haemolyticus*, yellow—*S. pasteuri*, dark blue—*S. simulans*, pink—*S. cohnii*, brown—*S. lugduensis*

## Data Availability

Due to privacy and ethical concerns, the data that support the findings of this study are available on request from the first author (U.K.).

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
