# Peer review of "Prevalence of Culturable Bacteria and Yeasts in the Nasopharynx Microbiota during the Physiological Course of Pregnancy"

_jcm, 2023, doi:10.3390/jcm12134447_

Round 1
Reviewer 1 Report (Previous Reviewer 1)
The work focuses on a very actual and important topic of the characteristics of culturable bacteria and yeasts in the nasopharynx of pregnant and non-pregnant women. Unfortunately, many research limitations made it impossible to obtain a full picture and conduct in-depth analyses. In addition, small and uneven research groups (PW and NPW), despite extensive statistical analyses, may make it difficult to draw the right conclusions.
The discussion is conducted in the correct way, although it includes a discussion of factors that were not covered by the study (e.g. the impact of SARS-CoV-2 infection on the nasopharyngeal microbiota). Also, the high variability of the CRP value in the course of physiological pregnancy makes the analysis of this parameter, in the context of the changing nasopharyngeal microbiota, very difficult.
Detailed comments:
Lines 157-158: provide the name of the company dealing with statistical calculations, currently in the manuscript there is an e-mail address
Lines 202-203: “Gram-positive bacteria colonized 50/53 (94.3%) of the PW cases and 27/30 (90.0%) of the NPW cases (p=0.463)”
According to the information provided in Table 2, in the NPW group, Gram-positive bacteria from the Staphylococcaceae and Streptococcaceae families were detected in 27 and 4 samples, respectively, which gives us a total of all tested samples in this group (30). Is this correct?
Supplementary Table S1 and elsewhere: standardize the spelling of Gram's staining; use Gram-positive/ negative in the whole manuscript
Discussion
Lines 362-368: the paragraph contains only study results, with no discussion of it
Lines 455-468: delete the paragraph
The manuscript has to be proofread by a native English speaker with expertise in the field.
Author Response
At the outset, we are very grateful to the Reviewers for their favourable view of our manuscript. We greatly appreciate their involvement and invaluable guidance. A more detailed response can be found in the appendix.

Reviewer 2 Report (New Reviewer)
The research as it is presented is sound, of general interest but not of huge medical significance. I was surprised more attention was not paid to Group A Streptococci, given the danger this group poses to pregnant women.
The introduction was fine, but there was too much discussion on the patterns of colonisation and less on what the overall significance of the lack of differences between pregnant and non-pregnant. The medical significance if any needs to be made clearer.
It would help to accept your text revisions before submitting your paper, as it negatively affected both the presentation and understanding of the science. The English phrasing needs also needs a re-check.
Author Response
At the outset, we are very grateful to the reviewers for their favourable view of our manuscript. We greatly appreciate their involvement and invaluable guidance. A more detailed response can be found in the appendix.

Reviewer 3 Report (New Reviewer)
A respectful greeting, I congratulate the authors for the presentation of their work, the idea worked on and the methodological design seems to me to be appropriate, it is an important question due to the potential for diagnostic and therapeutic applications that could be inferred, however I consider that the methodology does not allow us to infer that the condition of pregnancy could promote colonization by yeasts, the frequency of the finding does not allow us to affirm this and I consider that the conclusion should be modified, it is also important to make a little emphasis on the limitations of the results in the discussion, this study paves the way for other initiatives to assess this important question, with sample size being one of the most important considerations.
Author Response
At the outset, we are very grateful to the reviewers for their favourable view of our manuscript. We greatly appreciate their involvement and invaluable guidance. A more detailed response can be found in the appendix.

This manuscript is a resubmission of an earlier submission. The following is a list of the peer review reports and author responses from that submission.
Round 1
Reviewer 1 Report
General comments:
The actual manuscript reports a characterization of culturable nasopharyngeal microflora derived from pregnant and non-pregnant women. The issue is of importance in human medicine. Furthermore, the study provides the influence of predictor factors on developing the nasopharyngeal microflora in pregnant and non-pregnant women.
The manuscript is clear and well-written but contains too many tables. I think Tables 2, 4, 5 and 8 should be considered supplementary tables, as they cover only statistical calculations.
Detailed comments:
Abstract
Lines 24-25: According to the data presented in Table 8, the p˂0.05 value does not apply to Pseudomonadales.
Introduction
Lines 61-62: In the sentence “A better understanding of how the changes in the maternal respiratory microbiota may underlie..” there is only information about the respiratory microbiota, while the nasopharynx is the soft upper part of the throat that is a part of the GI tract. Needs a review.
Materials and methods
Lines 72-73: This sentence “This study was conducted to analyze whether pregnancy has an effect on the colonization of the nasopharyngeal microbiota (NPM)” appears to be more a study objective than a method.
Line 86: antibiotics only or antimicrobials?
Information about questionnaire data collected (age, place of residence, smoking, etc.) should be added to the “materials & methods” section. Add the physiological range of CRP for pregnant women.
Line 103: sheep blood, not ship’s blood
Lines 110-112: This sentence “In the majority of cases, uniform growth of bacterial colonies was observed, and sometimes some bacteria appeared predominant numbers” seems more as a result than as a method.
Results
Lines 142-143: repetition from subchapter 2.1, delete it
Line 183: unify the notation of % values to decimal values, here 90.0%
Add information on Gram-positive bacteria species that were isolated from mixed samples (with yeasts) in 2 samples from pregnant women.
Table 1
Place of residence: In the PW group there are only 52 samples (21 and 31), while n = 53?
Smoking: there are only 48 samples in the PW group (n=53) and only 24 samples in the NPW group (n=30)
Needs correction.
Figure 1
Since the groups are not numerically equivalent I suggest using percentages or presenting the data in a different type of chart, e.g. stacked column.
The numbers on the bars in Fig. 1 are not legible.
The sum of all cases in the PW group is 54 instead of 53 (12+21+7+6+5+3 = 54)
Table 3
The total % of Gram+, Gram- bacteria and yeasts of group PW equals 100.1%
Double parentheses at the 63.3 value.
Table 4
Marks such as ** and *** are not covered on Tab 4, therefore, remove them.
Table 7
Delete % at 11.3% value as this is covered in the column description.
Discussion
Lines 273-282: This paragraph is more a result than the part of the discussion
Lines 303-305: Use Past Simple
Line 325: Currently, data on
Line 347: There is a full stop after the word “strains”
Line 362: “Further research is needed” This sentence is very general and unclear. Further research is needed on what? Periodontal disease? Make it clearer.
Discuss the correlation between higher CRP levels and the microflora of the nasopharynx of pregnant women. At the moment, this information is scarce.
Line 392: Use Past Simple
Reviewer 2 Report
The authors studied the nasopharyngeal microbiota in the pregnant population and compared that with the normal healthy individual.
There are some comments for improvement:
1. Suggest to include sample size calculation in the methodology – to estimate the minimum number of sample size should be included to allow adequate power of the study.
2. Definition of “healthy” is not clear – please provide more details on the diseases that were being excluded in the study. What about those who are obese, with history of heavy alcohol consumption? Will these subjects be excluded, since different in lifestyle could potentially affect oral/nasopharyngeal microbiota?
3. The studied (pregnant) population with gestational age ranges from 5th up to 40th weeks. Would microbiota species be different in each period (trimester) of gestation? Of note this study recruited mainly pregnant women in the third trimester (n = 38) vs that in the first trimester (n = 5), which the types of microbiota detected could be skewed towards those in the late stage of the pregnancy. Suggest to include more subjects in the 1st and 2nd trimester.
4. Also, the heterogeneity in both the studied population and the control group, as the age in the NPW tends to be older. Suggest to include younger control group, with ratio studied population vs control
5. Methodology – suggest to include the timing of nasopharyngeal sample collection. Any special preparation/ conditions were needed for the patients before the nasopharyngeal sample being collected?
6. Table 1: suggest to elaborate the types of pets – since it may potentially affect the nasopharyngeal colonization with different exposure to the different types of pets.
7. Table 2 is unnecessary – suggest to combine the info with Table 1.
Round 2
Reviewer 2 Report
Except for the sample size, which is rather too small to represent the entire pregnant population, the rest of the responses by authors are acceptable.